# The way back home: The invisible burden of the emergency healthcare services

**Mustafa Enes Demirel** [ID][1]*, **Aysenur Ozcelik**[1], **Mustafa Bogan** [ID][2]

1 Emergency Department, Faculty of Medicine, Bolu Abant Izzet Baysal University, Bolu, Turkey,
2 Emergency Department, Faculty of Medicine, Düzce University, Düzce, Turkey

* mnsdmrl@hotmail.com

**Data Availability Statement:** All relevant data are within the paper and its Supporting Information files [Hometransprt (private-noname) SPSSfile. sav.].

## Abstract

Ambulance services around the world vary according to regional, cultural and socioeconomic conditions. Many countries apply different health policies locally. In Turkey, transportation from hospital to home has started to form an important part of ambulance services in recent years. The increase in the number of patients whose treatment has been completed and waiting to be referred may hinder the work of the emergency services. The aim of this study was to examine the costs, indications, and impact on workload of patients sent home by ambulance. Patients were divided into two groups according to the reasons for referral. The distance to home, transport time and cost were calculated according to the reasons for transport. Patients who were transferred to other clinics or hospitals by ambulance were excluded from the study. The findings showed that the hospital-to-home transfer rate during the study period was 11.4%. Although 9.7% of all cases transferred from our hospital to home were due to social indications, these cases accounted for 16.26% of the total costs. These results suggest that providing home transport services to selected patient groups for medical reasons should be seen as part of the treatment. However, the indications for home transport should not be exceeded and an additional burden should not be placed on the fragile health service.

## Introduction

Pre-hospital emergency healthcare services have been established to provide rapid treatment and to transport critical patients to the Emergency Department (ED) [1]. Throughout the world in general, emergency ambulance services are provided on a continuous 24-hour basis from provincial or regional control centres. Criteria for patient transfers are regulated with many national and international guidelines, and certain rules and regulations have been established [2, 3]. The provision of ambulance services and the rates of society using ambulances vary in developed and developing countries, depending on local, socioeconomic, and cultural conditions. With extensive changes to healthcare policies, the transformation of the healthcare program in Turkey aimed for the population to benefit from several services free of charge with only a small contribution to the social health insurance system [4]. With the increase in the elderly population of patients requiring care, the increase in demand has not been limited

**Funding:** The author(s) received no specific funding for this work.

**Competing interests:** The authors have declared that no competing interests exist.

to emergency ambulance services, but there has also been a noticeable increase in the need and demand for transfers from hospital to home [5]. This increase in demand for transfer from hospital to home has increased the number of patients in the ED who have finished treatment and are waiting for a transfer, and this has a negative effect on the availability of beds in ED and satisfaction [6].

The aim of this study was to determine the factors affecting the demand for ambulances for the transfer of patients from hospital to home from the ED of a tertiary hospital in the western Black Sea region of Turkey and to determine the indications, costs, and the workload on the emergency command system incurred by ambulance transfer from hospital to home.

## Methods

### Study design

This cross-sectional, retrospective study was conducted in a tertiary-level hospital in Turkey. The study protocol was approved by the Ethics Committee of Bolu Abant Izzet Baysal University (decision no: 2022/46, dated:08.03.2022).

### Data collection

The relevant data were obtained from the archives and statistics unit of the institution with the approval of the Ethics Committee. The data for research purposes were accessed on 24/05/2022. Information regarding ambulance transfers was obtained from the hospital information processing system and the data of the ambulance service affiliated with the provincial health directorate. During and after data collection, the authors had limited access to information that could identify individual participants, such as protocol number and demographic data.

### Ambulance service workflow

The non-hospital emergency medical services of the city are provided by the ambulance service under the Provincial Health Directorate. Two types of ambulances are available for emergency assistance: emergency medical ambulances and home transfer ambulances. Emergency medical ambulances are dispatched to patients who activate the emergency response system (dialing 112 in Turkey). These ambulances provide initial medical intervention to patients and ensure their transfer to appropriate hospitals within the relevant indications. There are a total of 47 emergency medical ambulances in the city ambulance service. These ambulances are white and red in colour and are equipped with all the necessary equipment for adult and pediatric patients, including transfer stretchers, oxygen sources, heaters, medical monitors, defibrillators, ventilators, delivery kits, burn kits, tracheostomy kits, essential medications for emergency interventions, trauma boards, cervical collars, and more. Each ambulance operates in a designated area within the city. In case of high demand, ambulances from outside the region may be assigned as needed.

Home transfer ambulances are used to transport patients who have completed their treatment in the hospital but cannot return home on their own for various reasons. These ambulances are white and blue in color and have more limited equipment. These include essential items such as transfer stretchers, oxygen sources, and heaters. They are not authorized to provide emergency medical care due to their limited equipment. There are two transfer ambulances in the city where the hospital is located, with one of the two ambulances located at our centre.

### Indications for transfer from hospital to home

Patients arriving at the hospital through the emergency response system may not always achieve complete recovery, and in some cases, they may be discharged with a need for home care. The transfer of such patients to their homes is made using home transfer ambulances in Turkey. However, when there is an insufficient number of these ambulances, emergency medical ambulances can also be utilized. The organization of patient transfers to their homes is coordinated through the hospital and the emergency response system. The referral criteria for home transfer ambulances are determined by national and local official healthcare institutions [7]. The following indications are provided for home transfer services to patients;

### Medical indications

Patients who have completed treatment in the emergency department and/or inpatient services but require home care, bedridden patients, disabled patients, patients requiring isolation (such as COVID-19 cases), and patients requiring oxygen support during the transfer (for patients already using oxygen devices at home).

### Social indications

Patients who have completed their treatment in the emergency department and/or inpatient services but who are without any relatives/caregivers or lack the financial means to return home.

### Measurement

Data were recorded of the patient age and gender, the indication for the home transfer, the time taken for the transfer and distance from the hospital, the time that the transfer was requested, the unit from which the patient was transferred, the reaction time of the command centre, and whether or not the ambulance took the duty. The reaction time was defined as the time from the recording on the emergency call system by a doctor to the time when the duty was assigned to an ambulance. The ambulances used for transfers were separated according to the area of duty as within the region or outside the region. The place to which the patients were taken at the end of the transfer was grouped as urban if a city neighborhood or town, or rural if a village, etc. The patients were separated into two groups according to the reason for transfer. Group1; patients transferred home due to medical indications, Group2; patients transferred home due to social indications. Patients who were transferred to other clinics or hospitals by ambulance were excluded from the study.

### Statistical analysis

Data obtained in the study were transferred to a computer and analyzed statistically using SPSS vn—23 software (SPSS Inc. Armonk, NY, USA). Descriptive statistics for continuous variables were expressed as the mean, standard deviation, minimum and maximum values. For variables that did not exhibit a normal distribution, the interquartile range (IQR) was used. Categorical variables were represented as counts (n) and percentages (%). The conformity of continuous variables to normal distribution was assessed with the Shapiro-Wilk test. The data of two groups were compared using the Mann Whitney U-test. The distribution of transfers by months was examined with the One-Sample Chi-square test. Relationships between two categorical variables were examined with the Pearson Chi-square test. A value of $p < 0.05$ was determined as the level of statistical significance.

**Table 1. Case statistics by year from the provincial ambulance service directorate.**

| Year | Hospital Emergency Department to home Cases–n (%) | Total Cases Transferred from Hospital to Home Cases–n (%) | Total number of cases |
|---|---|---|---|
| 2020 | 1,386(3.2%) | 4,859(11.4) | 42,462 |
| 2021 | 1,461(3.5%) | 4,788(11.6%) | 41,243 |
| 2022 (first 4 months) | 363(2.7%) | 1,356(10.1%) | 13,312 |
| Total | 3,210(3.3%) | 11,003(11.3%) | 97,017 |

## Results

In the study period of 2020–2022, a total of 3210 patient transfers were made by ambulance, as 1386 (43.2%) in 2020, 1461 (45.5%) in 2021, and 363 (11.3%) in the first 4 months of 2022 (Table 1). The time of the transfer request was examined in the time intervals of 00:00–07:59, 08:00–15:59, and 16:00–23:59 hours, and the transfer rates were determined to be 16.98% (n: 545) 35.58% (n: 1142), and 47.45% (n: 1523) respectively. The lowest rate of transfer was in the time interval of 00:00–07:59 and the highest rate was between 16:00–23:59 hours ($\chi^2 = 454.22$, p<0.001). Of the total transfers, 90.3% (n:2898) were because of a medical indication and the remainder were because of social indications (Table 2). The total cost of the transfers from hospital to home was calculated as 965,523 TL (Turkish lira). The cost for the transfer of patients with social indications was calculated to be 157,052 TL.

The median age of all the patients was 69 years (48–81). The patients transferred by ambulance because of social indications were determined to be significantly older than those transferred for medical indications (75 years (63–83) vs. 69 years (47–80); Z = 5.59, p<0.001). The patients comprised 1685 (52.49%) females, and the distribution of males and females in the groups transferred for medical or social indications were determined to be similar ($\chi^2 = 2.88$, p = 0.090). In respect of the time of transfers, the rate of transfer by ambulance because of social indication was lowest at 00:00–07:59 (n:35, 11.22%), higher at 08:00–15:59(n.24, 39.74%), and highest at 16:00–23:59 (n.153, 49.04%) ($\chi^2 = 8.63$, p = 0.013). Of the total transfers, 55.08% (n: 1768) were within the region, and transfers for patients with a medical indication were made at a higher rate within the region (n: 1628, 56.18%) ($\chi^2 = 14.55$, p<0.001) (Table 3).

The place of residence of the patients was urban in 69.78% (n:2240) of cases and the transfers because of medical indication were made at a higher rate for patients living in urban areas (n:2042, 70.46%) ($\chi^2 = 6.55$, p = 0.011). The median distance between the hospital and the patient's home was 10 km (7–52 km). The homes of patients transferred by ambulance because of social indications were determined to be at a greater distance (38 km (8–60.75 km)) than the homes of patients with medical indications (10 km (7–50 km)) (Z = -5,35, p<0.001). The median call reaction time was determined to be 16 secs (6–36 secs). No significant difference was determined between the call reaction times according to the reason for transfer (Z = -1.66, p = 0.098). The travelling time of the ambulance (from hospital to patient's home and return)

**Table 2. Transfers from hospital to home according to reasons.**

| | N | (%) |
|---|---|---|
| **Reason for Transfer** | | |
| Medical Indication | 2898 | 90.3 |
| Social Indication | 312 | 9.7 |
| Total | 3210 | 100 |

**Table 3. Characteristics of the patient transfers according to the transfer indication.**

| | Total | Reason for Transfer | | Z/$\chi^2$, p |
| --- | --- | --- | --- | --- |
| | | Medical indication | Social indication | |
| **Age—years,** median (IQR) | 69 (48–81) | 69 (47–80) | 75 (63–83) | -5.59, <0.001* |
| **Gender,** n(%) | | | | |
| Female | 1685 (52.49) | 1507 (52) | 178 (57.05) | 2.88, 0.090** |
| Male | 1525 (47.51) | 1391 (48) | 134 (42.95) | |
| **Time of transfer,** n(%) | | | | |
| 00:00–07:59 | 545 (16.98) | 510 (17.60) | 35 (11.22) | 8.63, 0.013** |
| 08:00–15:59 | 1142 (35.58) | 1018 (35.13) | 124 (39.74) | |
| 16:00–23:59 | 1523 (47.45) | 1370 (47.27) | 153 (49.04) | |
| **Region,** n(%) | | | | |
| Within the region | 1768 (55.08) | 1628 (56.18) | 140 (44.87) | 14.55, <0.001** |
| Outside the region | 1442 (44.92) | 1270 (43.82) | 172 (55.13) | |
| **Place of residence of the patient,** n(%) | | | | |
| Rural | 970 (30.22) | 856 (29.54) | 114 (36.54) | 6.55, 0.011** |
| Urban | 2240 (69.78) | 2042 (70.46) | 198 (63.46) | |
| **Hospital to home, km,** median (IQR) | 10 (7–52) | 10 (7–50) | 38 (8–60.75) | -5.35, <0.001* |
| **Reaction time, secs,** median (IQR) | 16 (6–36) | 16 (6–36) | 14 (6–33) | -1.66, 0.098* |
| **Ambulance travel time, min,** median (IQR) | 52.5 (35–101.55) | 50.64 (34.35–100.59) | 71.02 (41.71–106.31) | -4.04, <0.001* |
| **Total Cost, TL,**\*\*\* median (IQR) | 412 (272–529.25) | 406 (270–515) | 511.5 (384.25–584) | -9.27, <0.001* |

\* Mann-Whitney U test.

\*\*Pearson Chi-square test.

\*\*\* In 2020, 1 USD was 7.01 TL. In 2021, 1 USD was 8,861 TL. In 2022, 1 USD was 16.56 TL.

was median 52.5 mins (35–101.55 mins). The travelling time was determined to be significantly longer for patients transferred for social indications (71.02 mins (41.71–106.31 mins)) compared to those transferred for medical indications (50.64 mins (34.35–100.59 mins)) (Z = -4.04, p<0.001). The total cost of each transfer was calculated as median 412 TL (272–529.25 TL)). The transfer cost was determined to be significantly greater for patients transferred for social indications (511.5 TL (384.25–584 TL)) compared to those transferred for medical indications (406 TL (270–515 TL)) (Z = -9.27, p<0.001) (Table 3).

## Discussion

In this study, examinations were made of the indications of patients transferred by ambulance from the Emergency Department (ED) to home and the effect on the workload of the ED and the costs of healthcare services. That there have been insufficient reviews to date of transfers from hospital to home constitutes an important problem for EDs. According to the results of this study, the mean costs of transfers to home because of social indications were greater than those of patients transferred by ambulance because of medical indications. The reason for this was thought to be the higher cost due to the greater distance.

Emergency healthcare services are an extremely important public health service, with ambulances making the first intervention in life-threatening conditions and enabling patients to rapidly and safely reach the nearest ED [8]. However, the abuse of this important service by some people increases the pressure on ambulance services and prolongs the time to reach hospital for vital ambulance calls [9]. The ambulance services provided in developed and

developing countries and the rates of the populations using ambulances vary depending on local, socioeconomic, and cultural conditions. Data related to the healthcare system implemented are of guidance for other countries considering the implementation or not of different healthcare and social healthcare policies.

The worldwide increase, especially in industrialized countries, in the elderly population and the increase in chronic diseases and rapid technological developments have caused an increasing demand for emergency medical services, and a significant increase in the demand for and costs of ambulance services [10, 11]. Studies with large data clusters including useful information about patient demographic characteristics, treatment plans, and pre and post-hospital transfer and referrals will contribute to the literature and could guide social policies [12].

When the studies by Kıdak et al. and Zenginol et al. were evaluated, it was observed in our current research that the number of transfers to home had a higher percentage within total transfers in previous years and increased over the years [13, 14].

In respect of age and gender patient distribution, while the rate of male patients has been reported to be higher, the number of female patients in the group aged >65 years has exceeded the number of males [15]. Consistent with the literature, the current study results showed that the mean age was high and the proportion of female patients was greater. Age is a significant factor affecting the demand for emergency healthcare services, and there is a tendency for the elderly to more readily use ambulance services compared to younger patients [10]. The majority of the patients in the current study who were transferred from hospital to home were elderly patients. Our findings are similar to those of a study by Sultanoğlu et al. related to the demographic characteristics of patients transferred home [5]. The high mean age of the patients transferred by ambulance from hospital to home could be due to the more frequent presentations and greater morbidities of the elderly population, similar to previous findings in literature [15, 16].

Moreover, it was seen in the current study that the patients transferred home by ambulance because of social indications were older than the patients transferred because of medical indications. It seems to be inevitable that with the increasing proportion of the elderly in the total population, there will be an increase in the number of elderly patients presenting at ED, and therefore an increase in the numbers and costs of transfers from hospital to home for both medical and social reasons. Age is significant factor for demands on the system. The results of this study showed a greater need of the elderly population for transfer to home, and that there is insufficient standardization and number of studies in this area.

When the reasons for transfer by ambulance from hospital to home were examined in this study, it was seen that approximately one in ten transfers was due to social reasons. In the study by Sultanoğlu et al., transfers for social reasons were seen at a higher rate [5]. This was thought to be due to ambulance transfers because of COVID-19 in the current study during the pandemic. in the same study by Sultanoğlu et al., it was observed that the transfer time and average distance from hospital to home were similar to our findings [5]. Compared with the transfers for medical reasons, for the patients transferred for social reasons, the distance travelled, the duration of ambulance use, and the costs were statistically significantly greater. It can be predicted that an increased demand for home transfers for social reasons in the future will result in higher costs and a greater number of ambulances occupied in this way.

In the current study, the ambulance transfers to home were mostly in the evening (16:00–23:59), and due to the intense demand, ambulances from outside the region were used together with ambulances in the region of the hospital. This causes a deficit in the number of ambulances available in mass emergency situations. As there has been no study of the intensity of home transfers on emergency command call centres, no comparisons could be made.

The need for patient transfers to their homes is an expected situation. Patients who require home transfer are typically bedridden, in need of oxygen, require isolation, or face

transportation difficulties, making them unable to travel in a conventional passenger vehicle. In many examples it can be observed that as in Turkey, ambulances or ambulance services are used for these patients. This situation highlights that the issue is more global than local, with countries coming up with similar solutions [17, 18].

## Conclusion

The ambulance service should be seen as a part of treatment for specific groups with medical reasons to be able to return home after discharge from the ED. With the increasing elderly population in the future, the increased demand for home transfers for social reasons will increase the pressure on fragile healthcare systems by occupying a greater number of ambulances and increasing costs. There are insufficient studies and policies in this field. Therefore, there is a need for further collaborative multi-disciplinary studies and appropriate indications should be determined with regulations to standardise the transfer of patients from hospital to home by ambulance.

## Supporting information

**S1 Data.**
(SAV)

## Author Contributions

**Conceptualization:** Mustafa Enes Demirel, Aysenur Ozcelik, Mustafa Bogan.

**Data curation:** Mustafa Enes Demirel, Aysenur Ozcelik.

**Formal analysis:** Mustafa Enes Demirel, Aysenur Ozcelik, Mustafa Bogan.

**Funding acquisition:** Mustafa Enes Demirel, Aysenur Ozcelik, Mustafa Bogan.

**Investigation:** Mustafa Enes Demirel.

**Methodology:** Mustafa Enes Demirel, Mustafa Bogan.

**Project administration:** Mustafa Enes Demirel, Aysenur Ozcelik.

**Resources:** Mustafa Enes Demirel, Aysenur Ozcelik.

**Supervision:** Mustafa Enes Demirel, Aysenur Ozcelik, Mustafa Bogan.

**Validation:** Mustafa Enes Demirel, Mustafa Bogan.

**Visualization:** Mustafa Enes Demirel, Mustafa Bogan.

**Writing – original draft:** Mustafa Enes Demirel, Aysenur Ozcelik.

**Writing – review & editing:** Mustafa Enes Demirel, Mustafa Bogan.

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
