## [Decision Letter · Decision Letter 0]

15 Sep 2023

PONE-D-23-26422The way back to home: The invisible burden of the emergency healthcare servicesPLOS ONE

Dear Dr. Demirel,

Thank you for submitting your manuscript to PLOS ONE. After careful consideration, we feel that it has merit but does not fully meet PLOS ONE’s publication criteria as it currently stands. Therefore, we invite you to submit a revised version of the manuscript that addresses the points raised during the review process.

We look forward to receiving your revised manuscript.

Kind regards,

Burak Katipoğlu

Academic Editor

PLOS ONE

Journal Requirements:

5. We note you have included a table to which you do not refer in the text of your manuscript. Please ensure that you refer to Table 3 in your text; if accepted, production will need this reference to link the reader to the Table.

Additional Editor Comments:

Dear Authors,

There are some definitions that are not clear in the study.

Is there a separate ambulance system for home transport...

social indication, medical indication...etc

Therefore, the study needs to be re-evaluated after revision.

Kind Regards.

Reviewers' comments:

Reviewer's Responses to Questions

**Comments to the Author**

1. Is the manuscript technically sound, and do the data support the conclusions?

Reviewer #1: Yes

Reviewer #2: Partly

Reviewer #3: Partly

2. Has the statistical analysis been performed appropriately and rigorously? 

Reviewer #1: Yes

Reviewer #2: Yes

Reviewer #3: Yes

3. Have the authors made all data underlying the findings in their manuscript fully available?

Reviewer #1: Yes

Reviewer #2: Yes

Reviewer #3: No

4. Is the manuscript presented in an intelligible fashion and written in standard English?

Reviewer #1: Yes

Reviewer #2: Yes

Reviewer #3: Yes

5. Review Comments to the Author

Reviewer #1: If applicable, it would be beneficial to include a clear definition or specification of the criteria used to distinguish between social and medical indications in the methodology section. Providing such a definition would enhance the transparency of the study's patient classification.

Reviewer #2: 1. The authors supported ambulance transportation for medical causes from hospital to home for medical reasons, not social causes. Because ambulance use for social causes showed higher costs, and could lead to unnecessary healthcare system burden. Although I agree with the authors' opinion that ambulance use may be unnecessary for home transportation for social causes, there are lack of evidence to support the authors' suggestion in the paper. High costs are insufficient as the sole argument against ambulance use for social causes. Could the authors suggest the opposite opinion if ambulance transportation for medical causes required more costs? In my view, transportation due to medical causes also does not appear to be an emergency. If the authors want to suggest that unnecessary use of ambulances should be avoided, the authors must produce results to support this.

2. The authors should describe the emergency medical service system of Turkey in more detail in the study setting section. Is the ambulance mentioned in the paper a private or public ambulance? Does the public ambulance provide home transport in Turkey? In many countries, public ambulances do not provide interfacility or home transport. If the public ambulance provides home transport in Turkey, why are public ambulances responsible for transporting non-emergency patients? If it was private ambulances, what is the problem? There doesn't seem to be any problem if the patient has paid for transport. None of the bed-ridden, oxygen requirement and COVID-19 patients mentioned in the paper need to be transported urgently. There appears to be no compelling reason in the paper to expedite transportation due to medical reasons.

Reviewer #3: Dear Authors, Thank you for giving us the opportunity to review this article titled “The way back to home: The invisible burden of the emergency healthcare services”.

I don't have any conflict of interest.

I have some concern for this manuscript.

What is meant by 'effective use of ambulance services after medical treatment'? Are the personnel and ambulances that normally work for emergency patients working to take discharged patients home? If so, could an urgent patient have waited because there was no ambulance? This issue should be explained clearly. Or is there a different ambulance system that takes patients whose treatment is completed to their homes?

Is this a local problem in your country or an international problem? Explain with reference. If so, it may be more appropriate to evaluate it in a local journal.

Are private ambulances included in the study?

Your study lacked workload data.

Do you think it is necessary to compare your work with the pre-Covid period?

What is the medical and social indication? Explain with references.

Did the doctor decide whether patients had an indication? For what reasons was home transport by ambulance rejected and not performed?

Can you explain what reaction time is?

Transfer rates are evaluated according to hours. However, it needs to be compared with the application rate.

I have a few questions.

Decision making process after calling 112

Time from calling 112 to the arrival of the ambulance

Transportation process to the patient's home after calling 112

Did the patient go home with an ambulance coming from the hospital or with an ambulance coming from the station?

Please explain.

Is there a social indication definition for home transport in the literature? If not, could the reason why this is the first study conducted be because there is no social indication?

Have the ambulances used for transportation caused disruption in the 112 emergency service system?

This is the most important question and should be explained clearly.

If available, please include the country's health system legislation regarding the home transport system. And explain in detail the social and medical indications mentioned in this legislation.

I think your work needs revision.

Kind Regards.

6. PLOS authors have the option to publish the peer review history of their article (what does this mean?). If published, this will include your full peer review and any attached files.

Reviewer #1: No

Reviewer #2: No

Reviewer #3: No

---

## [Author Response · Author response to Decision Letter 0]

23 Nov 2023

Comments and Answers

* Edited

* Edited, certificate is attached.

* The dataset has been uploaded with data privacy considerations

* It was only mentioned in the Methods section. Another repeated phrase removed.

5. We note you have included a table to which you do not refer in the text of your manuscript. Please ensure that you refer to Table 3 in your text; if accepted, production will need this reference to link the reader to the Table.

* We fixed the phrase. 

6. Reviewer #1: If applicable, it would be beneficial to include a clear definition or specification of the criteria used to distinguish between social and medical indications in the methodology section. Providing such a definition would enhance the transparency of the study's patient classification.

* We have classified the indications for home transport into medical and social indications as stated in local and national rules.

7. Reviewer #2: 

a. 1. The authors supported ambulance transportation for medical causes from hospital to home for medical reasons, not social causes. Because ambulance use for social causes showed higher costs, and could lead to unnecessary healthcare system burden. Although I agree with the authors' opinion that ambulance use may be unnecessary for home transportation for social causes, there are lack of evidence to support the authors' suggestion in the paper. High costs are insufficient as the sole argument against ambulance use for social causes. Could the authors suggest the opposite opinion if ambulance transportation for medical causes required more costs? In my view, transportation due to medical causes also does not appear to be an emergency. If the authors want to suggest that unnecessary use of ambulances should be avoided, the authors must produce results to support this.

b. 2. The authors should describe the emergency medical service system of Turkey in more detail in the study setting section. Is the ambulance mentioned in the paper a private or public ambulance? Does the public ambulance provide home transport in Turkey? In many countries, public ambulances do not provide interfacility or home transport. If the public ambulance provides home transport in Turkey, why are public ambulances responsible for transporting non-emergency patients? If it was private ambulances, what is the problem? There doesn't seem to be any problem if the patient has paid for transport. None of the bed-ridden, oxygen requirement and COVID-19 patients mentioned in the paper need to be transported urgently. There appears to be no compelling reason in the paper to expedite transportation due to medical reasons.

* With the revision made to our article, social indications and medical indications have been elaborated in more detail. Home transfers based on social indications are not a problem unique to our country. In the reference examples we have added to the article, it is observed that home transfer criteria are similar in different countries. The issues related to home transfers are not local. However, the high number of patients and the fact that social indications are covered by social security constitute a significant part of the problem specific to our country. As stated in the conclusion section, despite the definition of separate vehicles for patient transfers, the need to resort to emergency ambulance services arises when a large number of transfers are required. This necessity is due to the high number of patients seeking emergency care and urgent need to free up beds in the emergency department. This situation makes the emergency health service fragile. In a society that is gradually aging and has an increasing number of individuals with chronic illnesses, we anticipate that patient transfers based on social indications will continue to rise in cost. We believe that this situation could disrupt the functioning of the ambulance service and lead to increasing costs.". 

8. Reviewer #3: Dear Authors, Thank you for giving us the opportunity to review this article titled “The way back to home: The invisible burden of the emergency healthcare services”.I don't have any conflict of interest.

I have some concern for this manuscript.

What is meant by 'effective use of ambulance services after medical treatment'? Are the personnel and ambulances that normally work for emergency patients working to take discharged patients home? If so, could an urgent patient have waited because there was no ambulance? This issue should be explained clearly. Or is there a different ambulance system that takes patients whose treatment is completed to their homes?

• With the revision made to our article, social indications and medical indications have been elaborated in more detail. In response to this question, the workflow of the Ambulance service is also summarized.

Is this a local problem in your country or an international problem? Explain with reference. If so, it may be more appropriate to evaluate it in a local journal.

• In the reference examples we added to the article, it can be seen that home transfer criteria are similar in some different countries. However, it is a condition that has not yet been mentioned in the literature.

Are private ambulances included in the study?

• Only institutional ambulances were evaluated in the study. Non-emergency transfers conducted with private ambulances were not included in our study because they were not recorded by the health directorate of the province from which we obtained the data. This phrase has also been added to the method section.

Your study lacked workload data.

• The study's main burden indicator is the proportion of patients moving home. (11.3%) It has increased 10 times compared to previous studies. It is thought that it will increase gradually.

Do you think it is necessary to compare your work with the pre-Covid period?

• In the methods section, medical and social indication criteria have been explained. The need for isolation due to COVID-19 is just one of these medical indications. The main purpose of our study is the patient and cost burden brought about by social indications. 

What is the medical and social indication? Explain with references.

• Medical indications: 

Patients who have completed treatment in the emergency department and/or inpatient services but require home care, bedridden patients, disabled patients, patients requiring isolation (such as COVID-19 cases), and patients requiring oxygen support during the transfer (for patients already using oxygen devices at home).

• Social indications: 

Patients who have completed their treatment in the emergency department and/or inpatient services but who are without any relatives/caregivers or lack the financial means to return home. However, it is a condition that has not yet been mentioned in the literature.

Did the doctor decide whether patients had an indication? For what reasons was home transport by ambulance rejected and not performed?

• It was decided by the ER doctor.

Can you explain what reaction time is?

• The reaction time was defined as the time from the recording on the emergency call system by a doctor to the time when the duty was assigned to an ambulance. This phrase has also been added to the method section.

Transfer rates are evaluated according to hours. However, it needs to be compared with the application rate.

• Since our study emphasized another closely related topic, statistical evaluations were simplified as much as possible.

I have a few questions.

Decision making process after calling 112

• It was decided by the ER doctor.

Time from calling 112 to the arrival of the ambulance

• An ambulance was dispatched to the nearest available unit by the emergency call center. We do not have information about the arrival time of the ambulance at our hospital. This phrase has also been added to the method section.

Transportation process to the patient's home after calling 112

• The ambulance assigned by the emergency call center (112) picks up the person to be transferred from the hospital, takes them home, and transports the patient to their place of residence. After dropping off the person, it proceeds to the relevant location for its next assignment or returns to its own ambulance station.

Did the patient go home with an ambulance coming from the hospital or with an ambulance coming from the station?

• The patient was sent with the ambulance that was available at that moment. First of all, patient transport ambulances located in the hospital are used. However, when it is very busy, emergency ambulances are also used. This phrase has also been added to the method section.

Please explain.

Is there a social indication definition for home transport in the literature? If not, could the reason why this is the first study conducted be because there is no social indication?

• We used this definition as an unnamed concept in the literature to better explain the topic we want to convey in our article. However, in many countries, it is a concept known as a transfer indication. With the revision we have made, we have also added links to articles related to these countries in our article (References 17 and 18)

17. Services BEH. In.: British Columbia, http://www.bcehs.ca/our-services/programs-services/patient-transfer-services

18. Health NMo. In.: New South Wales, https://www.health.nsw.gov.au/pts/Pages/info-for-patients.aspx

Have the ambulances used for transportation caused disruption in the 112 emergency service system?

• Emergency ambulances have also been used alongside transfer ambulances for patient transfers. No disruptions were reported during the course of the study due to this. However, especially the increasing home transfers associated with aging can render the emergency assistance system fragile, causing disruptions in the functioning of emergency services and delays in emergency assistance requests.

If available, please include the country's health system legislation regarding the home transport system. And explain in detail the social and medical indications mentioned in this legislation.

• In our healthcare system, there is no clear definition of social indications in the regulation governing transfer services. We used these concepts in our study on home transfers to provide a better explanation. We have also provided explanatory definitions of these terms in the methodology section. We have included official links to the legislation covering transfer services.

https://www.mevzuat.gov.tr/mevzuat?MevzuatNo=10834&MevzuatTur=7&MevzuatTertip=5

References 7; 

Turkey - Legal Gazette. In.: Turkey, https://www.resmigazete.gov.tr/eskiler/2013/09/20130920-8.htm.

---

## [Decision Letter · Decision Letter 1]

26 Jan 2024

PONE-D-23-26422R1The way back to home: The invisible burden of the emergency healthcare servicesPLOS ONE

Dear Dr. Demirel,

Thank you for submitting your manuscript to PLOS ONE. After careful consideration, we feel that it has merit but does not fully meet PLOS ONE’s publication criteria as it currently stands. Therefore, we invite you to submit a revised version of the manuscript that addresses the points raised during the review process.

We look forward to receiving your revised manuscript.

Kind regards,

Burak Katipoğlu

Academic Editor

PLOS ONE

Journal Requirements:

Reviewers' comments:

Reviewer's Responses to Questions

**Comments to the Author**

1. If the authors have adequately addressed your comments raised in a previous round of review and you feel that this manuscript is now acceptable for publication, you may indicate that here to bypass the “Comments to the Author” section, enter your conflict of interest statement in the “Confidential to Editor” section, and submit your "Accept" recommendation.

Reviewer #4: All comments have been addressed

Reviewer #5: All comments have been addressed

2. Is the manuscript technically sound, and do the data support the conclusions?

Reviewer #4: Yes

Reviewer #5: Yes

3. Has the statistical analysis been performed appropriately and rigorously? 

Reviewer #4: Yes

Reviewer #5: Yes

4. Have the authors made all data underlying the findings in their manuscript fully available?

Reviewer #4: Yes

Reviewer #5: Yes

5. Is the manuscript presented in an intelligible fashion and written in standard English?

Reviewer #4: Yes

Reviewer #5: Yes

6. Review Comments to the Author

Reviewer #4: Dear author

Thank for your inteesting to journal. My suggestions are below

1. Although you said you would give the results with the mean and standard deviation in the material section, you gave the results with the median ( percent 25-75%) in the results section. Please correct the relevant sentence in the method section.

Reviewer #5: 1. The cost analysis stated in the study should be stated in dollars, which is the international currency.

2. In the discussion section of the study, statistical data were used as numerical values. These data have already been mentioned in the results section. Numerical data from other articles should not be used. Instead, attention should be paid to the academic expression language expressing numerical values and corrected accordingly.

7. PLOS authors have the option to publish the peer review history of their article (what does this mean?). If published, this will include your full peer review and any attached files.

Reviewer #4: No

Reviewer #5: No

---

## [Author Response · Author response to Decision Letter 1]

1 Feb 2024

Reviewer #4: Dear author

Thank for your inteesting to journal. My suggestions are below

1. Although you said you would give the results with the mean and standard deviation in the material section, you gave the results with the median ( percent 25-75%) in the results section. Please correct the relevant sentence in the method section.

-We have added the necessary explanation;

For variables that did not exhibit a normal distribution, the interquartile range (IQR) was used.

Reviewer #5: 

1. The cost analysis stated in the study should be stated in dollars, which is the international currency.

Dear reviewer,

The exchange rate of the dollar in our country is not stable. However, healthcare service costs are fixed in Turkish lira. Due to the fluctuating USD-TL exchange rate, we believe that expressing the costs in USD for this study might lead to misinterpretation by the reader. Nevertheless, we aimed to provide readers with an insight in terms of dollars by presenting relevant information on the dollar exchange rates for the respective years below the table.

---

## [Editor Report · Decision Letter 2]

2 Feb 2024

The way back to home: The invisible burden of the emergency healthcare services

PONE-D-23-26422R2

Dear Dr. Demirel,

We’re pleased to inform you that your manuscript has been judged scientifically suitable for publication and will be formally accepted for publication once it meets all outstanding technical requirements.

Kind regards,

Burak Katipoğlu

Academic Editor

PLOS ONE
---

## [Editor Report · Acceptance letter]

26 Apr 2024

PONE-D-23-26422R2 

PLOS ONE

Dear Dr. Demirel, 

I'm pleased to inform you that your manuscript has been deemed suitable for publication in PLOS ONE. Congratulations! Your manuscript is now being handed over to our production team.

Kind regards, 

on behalf of

Dr. Burak Katipoğlu 

Academic Editor

PLOS ONE